# Measurement and Influencing Factors of Regional Economic Resilience in China

**Xinyu Zhang and Congying Tian \***

School of Economics, Shanghai University, Shanghai 200444, China; 787662785@shu.edu.cn
\* Correspondence: tcy0223@shu.edu.cn

**Abstract:** The COVID-19 outbreak in 2020 has underscored the paramount importance of regional economies' capacities to withstand and adapt to external shocks. Enhancing regional economic resilience and mitigating the adverse impacts on both the economy and society have emerged as critical imperatives for ensuring the sustainable development and transformation of the national economy. This paper employs an improved counterfactual method to measure the economic resilience index across 31 Chinese provinces and cities from 2001 to 2021, coupled with empirical analysis using a dynamic panel model to identify the influencing factors of regional economic resilience. Building upon this foundation, the study delves into the heterogeneous effects of various factors and different degrees of marketization on economic resilience across different regions. Research Findings: (1) There has been a significant improvement in the economic resilience levels of China's 31 provinces, with differences in economic resilience between regions far exceeding those in economic development levels, indicating substantial internal regional disparities. (2) Factors such as the marketization index, industrial structure, level of informatization, labor force size, labor quality, innovation capacity, and degree of government intervention all impact regional economic resilience and exhibit regional heterogeneity. Policy Recommendations: (1) It is crucial to address regional disparities while formulating regional development strategies and enhancing regional economic resilience. (2) Regions should accelerate market-oriented reforms, promote rational labor mobility, strengthen investment in human capital, foster innovation, and adjust the degree of intervention.

**Keywords:** national economy; sustainable development; counterfactual law; spatial evolution

## 1. Introduction

The COVID-19 pandemic, affecting over 200 countries, has highlighted the global reach and severe implications of health crises on public health, safety, and economic stability. This health emergency has severely threatened global public health, safety, and economic stability, marking a significant challenge to societal well-being and development. This challenge has intensified the global struggle against a variety of uncertain risks over the past two decades. Notably, the financial turmoil experienced in 2008 marked a pivotal moment, heralding deep changes in the global economy. In recent years, anti-globalization and trade protectionism has increased, weakening the forces driving global integration [1]. Against this backdrop, the regional response to special events and the ramifications of uncertainty, as well as strategies for recovery, have garnered significant attention from the academic community [2]. Subsequently, the study of regional resilience has gained momentum, mirroring the evolving global landscape. The nature of shocks, spanning from global financial crises such as the 2008 downturn to the widespread outbreak of COVID-19 in 2020, has become instrumental in deciphering the resilience of economies at the regional level. These seismic events have underscored the critical importance of understanding how regions respond to crises and navigate through uncertainty to achieve recovery and resilience. These events often arise suddenly and unpredictably, potentially leaving enduring impacts on economies [3]. In the contemporary context characterized by

complex international dynamics and uncertainties, enhancing regional economic resilience assumes paramount significance. Regional economic resilience, which refers to a region's ability to endure external crises and rebound through efficient resource allocation and economic structural enhancements, significantly influences the path of regional economic development in the foreseeable future. The ability to bolster resilience not only mitigates the adverse impacts on the economy and society but also promotes sustainable development, transforming and upgrading national economies. Consequently, it emerges as a crucial strategic imperative for national development agendas worldwide. Economic growth may stagnate or decline gradually, but sudden disruptions can cause immediate and severe consequences. Therefore, while the determinants of regional economic resilience evolve gradually over time, it is the immediate response to such shocks that truly reflects a region's resilience and shapes the trajectory of its economic development [4,5].

Economic resilience refers to an economy's ability to effectively respond to external disturbances, withstand shocks, and adapt its development trajectory [6]. Regional economic resilience, as the capacity of a region to withstand external crises and recover its economy through optimized resource allocation and upgraded economic structures, is crucial for shaping the future growth trajectory of regional economic development. In the current intricate international landscape marked by uncertainties, enhancing regional economic resilience and mitigating the adverse impacts on both the economy and society have become pivotal for ensuring sustainable development, transformation, and the upgrading of economies across various countries. This issue holds significant strategic importance for national development. Amidst China's economic uncertainties, building robust resilience is essential to manage strong shocks and navigate a volatile external environment [7]. In May 2020, The State Council of China highlighted in its Government Work Report that the Chinese economy has demonstrated strong resilience and immense potential. Empirical evidence since the outbreak of the epidemic further highlights the importance of prioritizing resilience-building efforts across various regions in China to effectively combat public emergencies [8]. Therefore, a comprehensive examination of regional economic resilience and its determinants, particularly focusing on China, not only provides valuable insights into the regions' capabilities to manage crises and risks but also offers fresh perspectives on addressing regional development disparities. Such analysis aims to inform the formulation of innovative macro-control policies grounded in both theoretical frameworks and empirical evidence.

The current literature primarily employs counterfactual methods of causal inference to measure regional economic resilience [9]. However, this approach, often benchmarked against the national economy, may not accurately reflect the nuances of China's regionally uneven economic development. Existing research on the factors influencing regional economic resilience primarily focuses on various aspects, such as industrial structure, innovation capacity, government governance, urbanization level, human capital, and economic openness [10–15]. Yet, there is a lack of fresh perspectives that adapt to evolving social dynamics in analyzing resilience determinants. Addressing these research gaps, this paper makes noteworthy contributions, particularly in the following areas. First, it enhances the causal prediction method within the counterfactual framework developed by Martin & Gardiner. This improved approach measures the economic resilience of diverse regions by considering regional growth rates. Unlike previous methods that relied solely on calculating economic resilience based on the national average, this approach considers the unique characteristics of China's regional disparities, providing a more comprehensive depiction of how various regions cope with shocks. Second, building upon existing research, this paper accentuates the role of structural factors while introducing two additional influential factors, namely, market reform and infrastructure, thus offering innovative insights. Finally, the study selects 31 provinces in China as its research subjects, providing fresh empirical evidence for understanding the measurement of regional economic resilience and its influencing factors.

## 2. Literature Review

The term "resilience", derived from the Latin "resilio", initially found its roots in the physical sciences and refers to a system's ability to return to its original state after a disturbance. However, it was Reggiani [16] who pioneered the adaptation of resilience within economic discourse. Reggiani defined economic resilience as the capacity of an economic system to maintain or restore its structural stability amidst external uncertainties within the realm of spatial dynamics. This conceptual migration into regional economics has since paved the way for a burgeoning field of study dedicated to examining the resilience of regional economies. Scholars like Foster [17] have highlighted that regional economic resilience involves a region's capacity for recovery and resistance against significant instabilities in the external environment. Hudson [18] and Martin and Sunley [9] have contributed to a more nuanced understanding of resilience, including aspects like vulnerability, resistance, robustness, and recovery capabilities, thus enriching the dialogue on how regional economies adapt to external shocks. Vulnerability refers to the sensitivity or predisposition of a regional economy to its growth structure before experiencing an impact. On the other hand, resistance signifies the degree of immediate response to shocks, influenced not only by the nature of the shock itself but also by the inherent characteristics of regional economic systems [19]. In essence, regional economic resilience embodies the ability of a regional economic system to rebound to its initial equilibrium state or evolve towards a more favorable developmental path through self-adjustment amidst external environmental changes. Current research on economic resilience centers on two main aspects.

The first aspect concerns the measurement of regional economic resilience. Martin [20] conducted a comparative analysis of regional economic resilience utilizing a sensitivity index. Empirical research has underscored that industrial structure stands out as the principal factor contributing to variations in regional economic resilience, emphasizing the dynamic nature of regional economic resilience itself. Briguglio et al. [21] were among the first to employ an indicator system to assess regional economic resilience. Their research revealed a strong correlation between economic resilience and the per capita GDP level of countries and regions. At present, the counterfactual method of causal inference is recognized by most scholars at home and abroad. Martin and Gardiner [22] analyzed the relationship among the resistance, decline, and resilience of 85 of the largest cities in the United Kingdom by making counterfactual predictions based on national responses. Based on Martin's calculation model, Liu et al. [23] considered the sign problem of positive and negative indicators and summarized a calculation formula with a wider application range. The nuanced characteristics of various shock scenarios warrant greater scrutiny, as the mechanisms involving structural and institutional factors may vary across different shock conditions. Additionally, it is imperative to account for the spatial correlation characteristics and spatial spillover effects influencing regional economic resilience [24]. In terms of systemic resilience, Douglas et al. [25] discussed the automatic association between disaster exposure and negative outcomes, highlighting the need to consider regions' positive responses and the sustainability of growth. The study conducted a comprehensive review of factors associated with vulnerability and resilience, employing a risk management framework to model the intricate interconnections among these variables and their implications for growth and distress outcomes. Wang and Wei [26] introduced a novel measurement approach for engineering resilience rooted in the theory of simple harmonic oscillation and institutional switches. They further explored the determinants of resilience within China's regional economies amidst the 2008 subprime mortgage crisis. Meanwhile, Davide [27] introduced a simplified framework for assessing structural engineering seismic resilience (SR), proposing a new recovery model to address the rapid economic recovery of regions from earthquake scenarios.

Secondly, the factors influencing regional economic resilience encompass various dimensions, including industrial structure, innovation capacity, government governance, urbanization, human capital, economic openness, and government intervention. Xu and Wang [12] utilized a binary logistic model to explore these factors, identifying that regions with a diverse range of unrelated industries were more adept at mitigating losses caused

by economic shocks. Simmie [10] provided insights into the impact of regional innovation systems on regional economic resilience in the UK, demonstrating that stronger regional innovation abilities correlated with heightened economic resilience. Zhang and Cui [14] examined the relationship between fiscal decentralization and economic resilience, highlighting the direct influence of government fiscal expenditure on economic resilience. Lester and Mai [11] emphasized the role of urbanization in promoting the exchange of regional commodities, thereby fostering greater integration of regional economies and enhancing economic resilience. Wang and Qiao [15] found a positive relationship between human capital and economic resilience, indicating that regions with higher levels of human capital were better equipped to withstand economic shocks. Wang and Guo [13] investigated the resilience of the electronic information industry, revealing that regions with higher levels of export trade tended to have more open regional environments and stronger global connections, contributing to enhanced regional industrial resilience. Government intervention and policy environments, as highlighted by Martin [3], are critical in navigating through external shocks, such as financial crises and natural disasters. Rios and Gianmoena [28] underscored the importance of regional government quality in enhancing resilience, with factors related to the learning and innovation atmosphere amplifying the significance of government quality.

In summary, a comprehensive review of both domestic and international literature reveals that while the counterfactual method is a preferred measurement technique, it often relies on national benchmarks, which may not suit the unbalanced economic development seen in regions like China. Thus, there is a need to adapt these methods to better reflect regional economic disparities. Based on the literature review and empirical evidence, the following hypothesis is proposed: an increase in the marketization index, industrial structure, level of informatization, labor force size, labor quality, innovation capacity, and the degree of government intervention contributes to enhancing regional economic resilience.

## 3. Methods and Data

### 3.1. Counterfactual Methods

At present, the measurement of economic resilience in academic circles mostly refers to Martin and Gardiner's counterfactual method, which has obtained a certain consensus. In mitigating the challenge posed by the intricate delineation of resistance and recovery periods, Liu et al., synthesized the measurement methodologies of economic resilience into a formula drawing from Martin's framework. This formula is articulated as follows:

$$\text{resil}_{i,t} = \frac{\Delta Y_i - \Delta E}{|\Delta E|}, \tag{1}$$

where $\text{res}_{i,t}$ is the relative economic resilience of region i in year t, and $\Delta Y_i$ is the actual economic change of region i, see Formula (2). Given that $\Delta E$ is based on the overall economic growth of the country (or province) where region i is located, the expected economic change of region i can be predicted, see Formula (3).

$$\Delta Y_i = Y_{i,t} - Y_{i,t-k} \tag{2}$$

$$\Delta E = \frac{Y_{r,t} - Y_{r,t-k}}{Y_{r,t-k}} \cdot Y_{i,t-k} \tag{3}$$

The projection for a region unaffected by external disruptions relies on the overall economic growth trajectory of the nation. However, the counterfactual framework by Martin and Gardiner requires using national responses to shocks as a baseline for cross-regional comparison, which assumes uniform responses across all regions. Yet, empirical evidence from China, particularly during significant events like the 2008 financial crisis and the 2020 COVID-19 pandemic, reveals substantial regional performance variances, as illustrated in Figure 1. Additionally, factors such as geographic location and economic foundation contribute to the well-documented imbalanced development among China's

regions. Consequently, forecasting the resilience of unaffected regions should not rely solely on the average level of the national economy. Regional disparities necessitate tailored approaches to resilience building that account for the unique characteristics and challenges faced by each region.

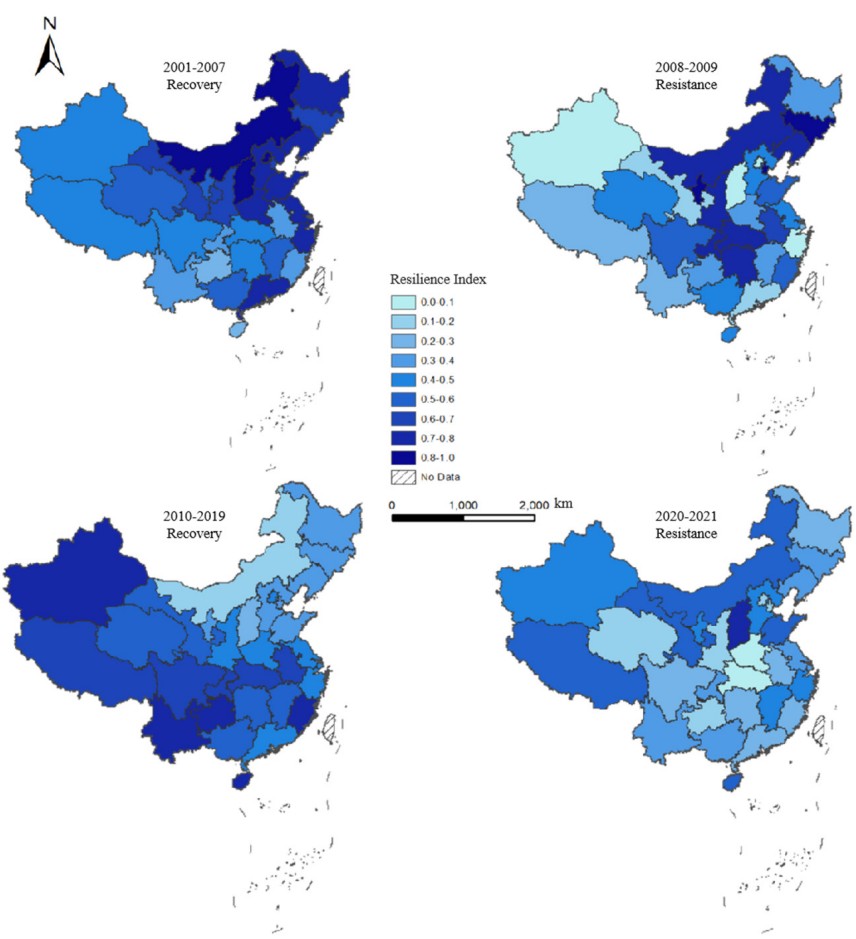

**Figure 1.** Spatial evolution of regional economic resilience in 31 provinces and cities of China (regional average growth rate). Approval No. GS(2019)1815.

Martin and Gardiner also mentioned several solutions to the problem of reasonable norms of counterfactual forecasting, one of which is to use a statistical time series model or some appropriate structural model to predict the growth path before the shock. Time series models, such as Autoregressive Moving Average (ARMA), Autoregressive Integrated Moving Average (ARIMA), Seasonal ARIMA (SARIMA), and Exponential Smoothing, are traditional tools for analyzing trends in time series data. These models can capture the cyclical, trend, and random components in the data, thus providing a reasonable prediction of future data changes. On the other hand, structural models are developed based on domain-specific theories and knowledge, incorporating systematic factors from economic, sociological, or physical principles to explain data variations and forecast the potential impacts of various influencing factors on the system. While statistical models analyze historical data patterns, structural models provide a theoretical basis for understanding systematic influences. Thus, in counterfactual scenarios, both methods offer valuable insights into predicting growth paths.

In light of the unique aspects of China's regional economic development, this paper proposes specific modifications to the resilience calculation method, divided into two approaches (as the paper aims to compute annual economic resilience for each region, the value of k is set to 1).

First, the regional average growth rate is used to measure the expected regional change rate.

$$\Delta\overline{G} = G_{\overline{i}}\cdot Y_{i,t-1} \tag{4}$$

Here, $G_{\overline{i}}$ is the arithmetic average of the economic growth rate of region i during the year under observation, and $\Delta\overline{G}$ is the expected change in the economic growth rate of region i in year t. Then, the improved Formula (5) for the calculation of regional economic resilience in this paper is obtained:

$$resil_{i,t} = \frac{\Delta Y_{i,t} - \Delta\overline{G}}{|\Delta\overline{G}|} \tag{5}$$

where $\Delta Y_{i,t}$ is the actual change value of the economic operation of region i in year t, and $res_{i,t}$ is the economic resilience of region i in year t. According to Formula (5), the central value of regional economic resilience is 0. When the resilience value is positive, the region is more resistant to external shocks, and the economy is more resilient, and vice versa.

Second, the expected rate of change in the region is measured by constructing a regional forecast model.

$$Y_{i,t} = dummyF + dummyC + \varepsilon_{i,t} \tag{6}$$

The financial crisis in 2008 and the COVID-19 epidemic in 2020 are set as dummy variables dummyF and dummyC respectively. If the impact occurs in the current year, dummy = 1, and if the impact occurs in the other year, the dummy = 0 (considering the sustained impact of the impact here, the region is considered to be affected by the impact in the year and after the impact). After individual fixation, the residual term $e_{i,t}$ is obtained by regression, and then the expected growth rate of economic operation in region i in year t is calculated as follows:

$$Eg_{i,t} = (e_{i,t} - e_{i,t-1})/|e_{i,t-1}| \tag{7}$$

where $e_{i,t}$ is the expected economic level of region i in year t after controlling the 2008 financial crisis, the impact of COVID-19 in 2020, and regional characteristics, and $Eg_{i,t}$ is the expected economic growth rate of region i in year t. Thus, the expected change value of economic operation of region i in year t can be obtained as follows:

$$\Delta E_G = Eg_{i,t}*Y_{i,t-1} \tag{8}$$

Then, the improved Formula (2) for the calculation of regional economic resilience in this paper is obtained:

$$resil_{i,t} = \frac{\Delta Y_{i,t} - \Delta E_G}{|\Delta E_G|} \tag{9}$$

where $\Delta Y_{i,t}$ is the actual change value of the economic operation of region i in year t, and $res_{i,t}$ is the economic resilience of region i in year t. According to Formula (9), the central value of regional economic resilience is 0. When the resilience value is positive, the region is more resistant to external shocks, and the economy is more resilient, and vice versa.

### 3.2. Model Setting

Through the construction of the OLS regression model, this paper empirically analyzes the influencing factors of regional economic resilience using the following formula:

$$resil_{i,t} = \alpha_0 + \alpha_1 market_{i,t} + \alpha_2 IR_{i,t} + \alpha_3 infor_{i,t} + \alpha_4 labor_{i,t} + \alpha_5 illit_{i,t} + \alpha_6 patent_{i,t} + \alpha_7 Gov_{i,t} + \mu_i + \varepsilon_{i,t} \tag{10}$$

where i represents each province, t represents the year, $resil_{i,t}$ represents the regional economic resilience index, $market_{i,t}$ represents the marketization index, $IR_{i,t}$ represents the industrial structure, $infor_{i,t}$ represents the informatization level, $labor_{i,t}$ represents the size of the labor force, $illit_{i,t}$ represents the quality of the labor force, $patent_{i,t}$ represents the

innovation ability, $\text{Gov}_{i,t}$ represents degree of government intervention, $\alpha_0$ is the intercept term, $\alpha_1, \alpha_2 \cdots\cdots \alpha_7$ represents the regression coefficient of each explanatory variable, $\mu_i$ is the regional heterogeneity term, and $\varepsilon_{i,t}$ is the random interference term.

*3.3. Variable Description*

3.3.1. Dependent Variable

Regional economic resilience index (Resil): The improved counterfactual method is used to measure the resilience level of each province. The greater the value, the greater the economic resilience. For the specific calculation process, see the measurement of regional economic resilience in the above section.

3.3.2. Independent Variables

The descriptive statistics of selected variables are shown in Table 1.

**Table 1.** The descriptive statistics of variables.

| Variables | Characterization | Obs | Mean | Std | Min | Max |
|---|---|---|---|---|---|---|
| Resil | Regional economic resilience index | 372 | 0 | 0.550 | −1.603 | 1.987 |
| Market | Comprehensive marketization index | 372 | 7.454 | 2.130 | −0.160 | 11.490 |
| IR | The proportion of the output value of the tertiary industry | 372 | 0.924 | 0.310 | 0.191 | 1.897 |
| Infor | The ratio of the number of mobile phone users to the resident population at the end of the year | 372 | 0.891 | 0.283 | 0.280 | 1.878 |
| Labor | Population aged 15–64 | 372 | 2.933 | 1.979 | 0.183 | 12.884 |
| Illit | The proportion of the population aged 15 and over who are illiterate | 372 | 6.236 | 6.164 | 0.890 | 41.194 |
| Patent | Number of regional patent applications granted | 372 | 4.745 | 8.047 | 0.009 | 70.973 |
| Gov | Regional government expenditure as a percentage of GDP | 372 | 0.285 | 0.203 | 0.100 | 1.354 |

Marketization index (Market): There are many methods used to measure the degree of marketization, but a single indicator can only reflect one aspect of marketization. Therefore, a comprehensive marketization index can fully reflect the level of marketization in a region. Thus, this paper uses the comprehensive marketization index estimated by Fan et al. [29] to measure the regional marketization degree.

Industrial structure (IR): Generally, the literature adopts the non-agricultural output ratio as a measure of industrial structure advancement based on the Clark–Fisher theorem. Nonetheless, this approach overlooks the dynamic shifts in economic composition. To address this limitation, this study follows the methodology of Gan et al. [30] by utilizing the proportion of output value generated by the secondary and tertiary sectors as an indicator of industrial structure upgrading.

Information level (Infor): Wang and Yu [31] used the Information Technology Composite Index (CIIC) to measure the level of information technology in China. However, since the CIIC involved many indicators and the data were limited, this paper draws on the research of Chen et al. [32] and uses one of the indicators to measure the penetration rate of mobile phones.

Labor force size (Labor): According to international general standards, individuals 15 to 64 years old belong to the range of working age. Therefore, this paper uses the population aged 15–64 to measure the size of the labor force in each region.

Labor force quality (Illit): Labor force quality is typically evaluated using two key metrics: per capita years of education and the illiteracy rate. The illiteracy rate serves as a structural indicator, reflecting the extent of compulsory education, while the metric of per capita years of education functions as an intensity indicator, gauging the overall

educational attainment level of the population [33]. In this context, the illiteracy rate is particularly policy oriented, offering insights into the effectiveness of educational policies and the prevalence of basic literacy skills within the labor force. A lower illiteracy rate signifies higher labor quality, indicating a more educated and skilled workforce.

Innovation ability (Patent): The most direct manifestation of regional innovation lies in the successful implementation of scientific and technological R&D activities, which promotes the progress of social technology and the improvement of labor productivity, benefiting the region's ability to resist external crises. Therefore, this paper uses the number of regional patent grants to measure regional innovation capacity.

Government intervention level (GOV): This study quantifies the extent of government intervention by assessing the ratio of government financial expenditure to the GDP of the region.

### 3.4. Data

This study examines the economic resilience of 31 provinces and cities in China from 2001 to 2021, focusing on data primarily from 2008 to 2020, due to its availability. The analysis explores the factors influencing economic resilience, utilizing marketization index data from Fan and Wang's latest "Report on China's Marketization Index by Province (2021)". Labor force data for 2010 and 2020 were obtained from the sixth and seventh Chinese population censuses, respectively, with data for the intervening years extrapolated from population sampling surveys. Additional indicators were sourced from the statistical yearbooks of the 31 provinces and cities, and map vector data were accessed from the Earth System Science Institute's Data Sharing Platform (http://www.geodata.cn) accessed on 11 November 2022.

## 4. Results

### 4.1. Spatial Evolution of China's Regional Economic Resilience

This paper, based on the GDP data of 31 provinces and cities from 2001 to 2021, measures and compares the levels of economic resilience across different regions in China using two improved methods. Considering that most economic recessions of regional systems after external shocks only last one year, this paper defines the year and the second year of the impact as the resistance period of the regional economy, and the recovery period of the regional economy begins in the third year after the impact. The division of the period is basically the same as that of other scholars [34]. The natural interval classification method of ArcGIS10.8 software was adopted, and the data standardization range was [0, 1]. The greater the toughness value, the stronger the impact on coping ability. After calculating the economic resilience values for each province and city in China, the arithmetic averages within the year range for different resistance and recovery periods are presented in Figures 1 and 2.

From Figure 1, it is observed that the period from 2001 to 2007 marked a rapid recovery phase following the Asian financial crisis. After joining the WTO, China experienced an economic boom, with strong resilience in the eastern coastal regions and a robust development momentum in the northeastern region. Some western areas, such as Guangxi, Gansu, and Shanxi, also showed significant improvement, but the issue of the "central collapse" became more pronounced. During 2008–2009, regions such as Beijing, Shanghai, and Guangdong, with well-developed financial sectors and high economic openness, faced significant challenges due to the 2008 financial crisis. Most eastern coastal and some central and western provinces suffered severe impacts. From 2010 to 2019, China experienced a gradual economic recovery following the aftermath of the financial crisis, with the growth rates in the central and western regions outpacing those in the eastern regions. These regions demonstrated faster economic rebound and notably enhanced resilience, while the eastern and northeastern regions exhibited a slower pace of economic development, focusing on maintaining stability. Across China, each province formed a tightly interconnected and organized hierarchy of economic resilience, with Jiangsu, Shandong, Guangdong, Hubei, and Shanxi emerging as pivotal clustering points and

influential centers within the spatial correlation framework of economic resilience [35]. The years 2020 and 2021 were a period of resistance to the global COVID-19 pandemic, with Hubei, Henan, Beijing, and Tianjin among the regions most severely impacted by the outbreak. The resilience values of most central and western regions significantly declined, indicating a substantial impact, while the eastern regions stabilized the pandemic situation earlier, minimizing the negative impacts of the outbreak on the economy and society to the greatest extent [36]. Overall, external shocks have exacerbated the existing issues of unbalanced and uncoordinated regional development in China. Consequently, there is an imperative need to delve deeper into the factors that shape regional economic resilience. This exploration is vital to formulate pragmatic policy recommendations aimed at fostering stable economic development across diverse regions in China.

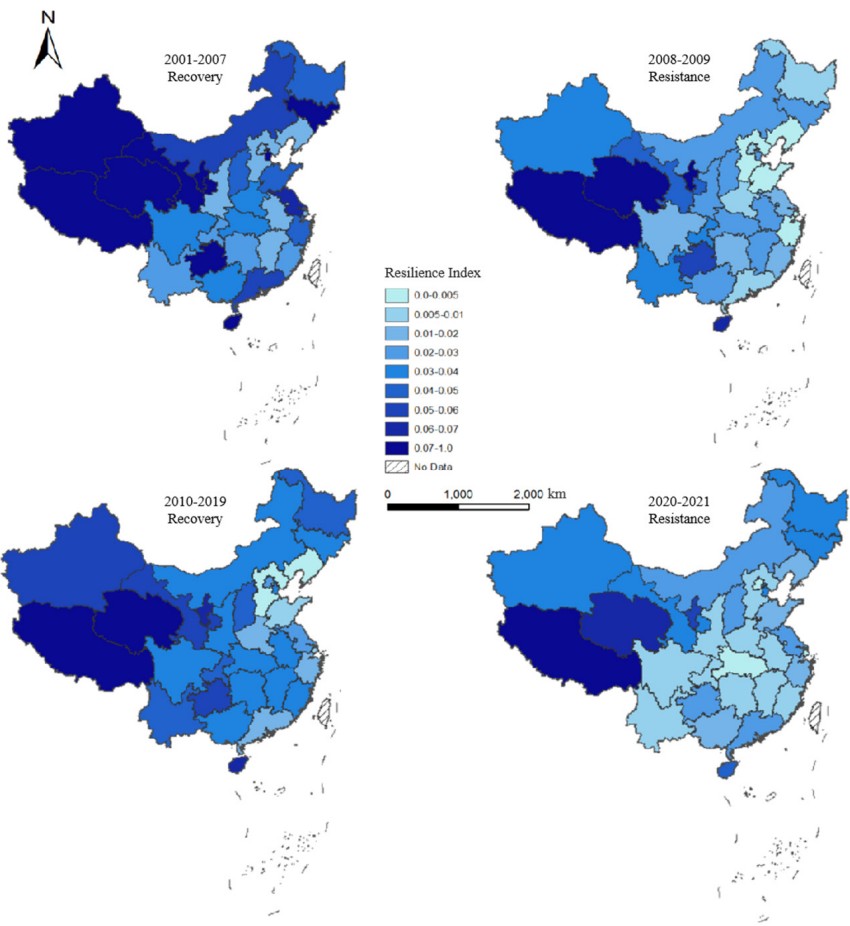

**Figure 2.** Spatial evolution of regional economic resilience in 31 provinces and cities of China (regional forecasting model). Approval No. GS(2019)1815.

Secondly, the economic resilience level of 31 provinces and cities in China in the four periods calculated based on the regional forecast model is shown in Figure 2. To facilitate comparison and visual display, the data is standardized in the range of [0, 1]. The greater the resilience value, the stronger the ability to cope with shocks. The period of 2001–2007 was the golden period of economic recovery after China's accession to the WTO. The development momentum was good, and the resilience level was relatively high. The eastern coast and most of the western regions such as Tibet, Qinghai, and Ningxia exhibited strong resilience. The economy of the northeast region was also greatly improved, but the development of the central region was slow. Thus, the gap between regions began to widen. Judging from the resistance period of the financial crisis in 2008–2009, Shanghai, Zhejiang, and Guangdong were severely affected, and most of the eastern coastal and northeastern regions were greatly affected. During the recovery period spanning from 2010 to 2019, the

central and most western regions exhibited high resilience levels, leading to rapid regional economic recovery. Conversely, the eastern and northeastern regions exhibited a tendency towards stability, characterized by economic growth rates slower than those observed in the central and western regions. However, an analysis of regional average growth rates indicates that the overall recovery of the eastern region was more robust. In 2020–2021, the global community faced the repercussions of the COVID-19 epidemic, with Hubei, Beijing, and Shanghai emerging as the epicenters of the crisis. Consequently, the resilience values of the central and most western regions experienced a significant decrease, rendering them more susceptible to the impact of the epidemic. In contrast, the eastern region managed to control the spread of the epidemic earlier, resulting in higher resilience levels across most provinces and cities compared to the central and western regions. Generally, regions with higher infection rates were more adversely affected, but even the less-infected western region saw a significant decrease in resilience.

Finally, a comparative analysis contrasting the outcomes from regional average growth rates (illustrated in Figure 1) with those obtained through the regional forecast model (depicted in Figure 2) shows that the former approach more effectively captures the nuances of GDP growth rate changes. The calculation based on regional average growth rates demonstrates more stable and elevated economic development in the eastern coastal regions, with less volatility in resilience levels compared to the central, western, and northeastern regions. This method aligns more closely with the actual socio-economic progression observed in China. The analysis underscores the exacerbated issue of the imbalance and lack of coordination of regional development in China due to external shocks, emphasizing the necessity for a deeper examination of the factors influencing regional economic resilience. Such insights are crucial to developing targeted policy recommendations aimed at fostering consistent regional economic development across the nation.

### 4.2. Multicollinearity Test

To mitigate the potential issue of multicollinearity between independent variables in regression analysis, this paper initially computes the Pearson correlation coefficient between pairs of independent variables for preliminary assessment. Typically, a correlation coefficient exceeding 0.8 suggests a strong correlation, indicating a more serious collinearity problem. The test results, as depicted in Table 2, reveal that the correlation coefficient between "illit" and "market" is the strongest at −0.625. However, the absolute values of correlation coefficients between other pairs of independent variables are all below 0.8, signifying the absence of severe multicollinearity among the independent variables.

**Table 2.** Test of correlation coefficient of independent variables.

| Variables | Market | IR | Infor | Labor | Ill | Patent | Gov |
|---|---|---|---|---|---|---|---|
| Market | 1.000 | | | | | | |
| IR | −0.009 | 1.000 | | | | | |
| Infor | 0.445 | −0.601 | 1.000 | | | | |
| Labor | 0.471 | 0.352 | −0.080 | 1.000 | | | |
| Illit | −0.625 | −0.115 | −0.290 | −0.265 | 1.000 | | |
| Patent | 0.602 | −0.109 | 0.430 | 0.504 | −0.190 | 1.000 | |
| Gov | −0.743 | −0.319 | −0.066 | −0.507 | −0.832 | 0.306 | 1.000 |

The paper employs the Variance Inflation Factor (VIF) to assess the model's overall correlation, setting a threshold value of 10 to mitigate serious collinearity concerns among the independent variables. The findings, indicating VIF values below 10 (test results are shown in Table 3), affirm the model's integrity for estimation.

**Table 3.** Independent variable VIF test.

| Variables | VIF | 1/VIF |
|---|---|---|
| Market | 4.43 | 0.226 |
| IR | 1.98 | 0.505 |
| Infor | 3.17 | 0.316 |
| Labor | 1.26 | 0.793 |
| Illit | 5.12 | 0.195 |
| Patent | 1.99 | 0.502 |
| Gov | 8.12 | 0.123 |
| Mean VIF | 3.72 | |

*4.3. Baseline Estimation Results*

In this paper, the fixed effect model was confirmed using the Hausman test. Models (1)–(5) in Table 4 are the regression results of gradually increasing explanatory variables, and model (6) in the last column is the final regression result.

**Table 4.** Results of benchmark regression.

| Variables | (1) Resil | (2) Resil | (3) Resil | (4) Resil | (5) Resil | (6) Resil | (7) Resil |
|---|---|---|---|---|---|---|---|
| Market | 0.359 *** (0.035) | 0.164 *** (0.033) | 0.099 ** (0.039) | 0.082 ** (0.039) | 0.076 * (0.040) | 0.088 ** (0.040) | 0.103 ** (0.041) |
| IR | | −1.176 *** (0.178) | −0.878 *** (0.212) | −0.882 *** (0.208) | −0.802 *** (0.216) | −0.844 *** (0.221) | −0.833 *** (0.221) |
| Infor | | | 0.580 *** (0.177) | 0.631 *** (0.178) | 0.835 *** (0.198) | 0.900 *** (0.199) | 0.664 *** (0.201) |
| Labor | | | | 0.083 *** (0.028) | 0.085 *** (0.028) | 0.085 *** (0.028) | 0.074 *** (0.027) |
| Illit | | | | | −0.045 *** (0.016) | −0.044 *** (0.016) | −0.041 ** (0.018) |
| Patent | | | | | | 0.014 ** (0.006) | 0.012 ** (0.006) |
| Gov | | | | | | | 1.728 ** (0.761) |
| Constant | −3.326 (0.323) | −1.278 *** (0.334) | −1.638 *** (0.360) | −1.681 *** (0.359) | −2.036 *** (0.408) | −2.134 *** (0.403) | −2.255 *** (0.438) |
| Obs | 372 | 372 | 372 | 372 | 372 | 372 | 372 |
| $R^2$ | 0.230 | 0.346 | 0.368 | 0.382 | 0.397 | 0.403 | 0.413 |

Note: Robust standard error is reported in parentheses, *** $p < 0.01$, ** $p < 0.05$, * $p < 0.1$.

The regression analysis reveals that the marketization index, a key indicator of marketization extent, has a coefficient of 0.088 and is significant at the 5% level. This signifies that fostering marketization construction can notably enhance the resilience level of the regional economy. As marketization increases, capital is more efficiently reallocated from less efficient to more efficient sectors, improving regional capital allocation efficiency [37]. According to the resilient adaptive cycle theory, a robust market structure can swiftly redirect resources to sectors less impacted by shocks, thereby mitigating the overall market system damage.

The regression analysis yielded a coefficient of −0.844 for the industrial structure variable, exceeding the 1% significance level, indicating that regions with rapidly evolving industrial structures are more resilient during crises. Unlike industries characterized by strong specialization, the service sector inherently embodies attributes such as intangibility, diversity, and concurrent production-consumption processes. When such specialized economies face external shocks, the impact spreads to interconnected industries, increasing regional risk vulnerability. A diversified industrial framework serves to effectively attenuate the impact risks by dispersing them across various sectors, thereby mitigating

the prevalence of austerity measures and large-scale layoffs. Additionally, it enables a quicker resumption of operations and production after a crisis, aiding rapid economic recovery. This diversification not only bolsters regional resistance to adverse impacts but also contributes, to a certain extent, to enhancing overall resilience [38].

The regression coefficient of the informatization level stands at 0.900, passing the significance level test at 1%, signifying that augmenting the informatization level substantially bolsters regional economic resilience. Advanced levels of informatization play a pivotal role in mitigating information costs when a region faces external shocks. Effective communication infrastructure is crucial for mitigating economic and social losses during financial crises or health emergencies, such as the COVID-19 epidemic. Consequently, the region's capacity to withstand external shocks experiences a notable enhancement. Numerous cities across China are actively pursuing the development of smart cities, leveraging information technology to integrate urban resources and data. This concerted effort holds significant promise in elevating the level of economic resilience. By enhancing urban operational efficiency and bolstering residents' quality of life, the advancement of smart cities contributes to fostering sustainable development and fortifying the resilience of regional economies [39,40].

Labor force analysis shows significant findings. The size (0.085) and quality ($-0.044$) of the labor force both show statistical significance at the 1% level, highlighting a positive correlation between the labor force's size and quality and regional economic resilience. The crucial role of the labor force in regional development is particularly evident during external shocks, where a robust working-age population is vital for resuming business operations and production, thus facilitating rapid economic recovery. However, according to China's seventh national census, the country's working-age population has dropped to 880 million. In addition, the median age of the labor force has reached 38 years old, and the labor force is gradually getting older. In this context, the labor force population should not only improve in quantity but also improve in quality, to achieve the effect of "quality supplement quantity". On the one hand, the improvement in labor quality will enhance the knowledge spillover of human capital, enhance regional innovation ability, and have strong externalities [41]. On the other hand, the improvement in labor quality can match more emerging industries, improve labor productivity, and create a new growth path over time, which is conducive to the region's continuous adaptation to the external environment to restore stable development.

The regression coefficient of innovation ability stands at 0.014, with statistical significance at the 5% level, suggesting that regions with higher innovation capacities better handle external crises. Generally, most regions are susceptible to the path dependence effect during external crises, wherein negative path dependence undermines regional economic resilience and often precipitates regional economic recession. Innovation is conducive to the region to break the traditional path dependence through the adjustment of industrial structure to enhance the ability to cope with the crisis.

The regression coefficient for government intervention stands at 1.728, with a significant impact at the 5% level. In response to the COVID-19 outbreak, Chinese local authorities swiftly enacted various measures, including medical aid, supply chain logistics, and public safety initiatives, to contain the spread of the virus. However, these interventions had a temporary dual impact on economic resilience, as noted by Swanstorm [42]. While interventions aimed at safeguarding public health may initially disrupt economic and social functions, such effects are transient, especially given the gradual containment of COVID-19 in China since 2020. Therefore, reinstating pre-pandemic governmental interventions could expedite the recovery of economic and social activities to their pre-crisis levels.

*4.4. Robustness Test*

To test the robustness of the baseline regression, this section uses regional economic resilience based on the national average growth rate measure (Resil_N) and regional forecasting model measure (Resil_D) as alternative indicators of the explained variables

and uses per capita years of education (Edu) as the surrogate indicators of labor force quality (Iillit). In Table 5, models (1)–(2) are regression results replacing explained variables, and model (3) is regression results replacing explained variables. Upon substitution of the explanatory variables, it becomes evident that the magnitude of the labor force size in model (1) lacks statistical significance, while the labor force quality in model (2) similarly fails to exhibit significance. However, the significance levels and directional indicators of other influencing factors remain congruent with the baseline regression outcomes, indicating a degree of robustness in the baseline model. Additionally, the metric of per capita years of schooling in model (3) shows a significant positive impact on regional economic resilience. This result supports the robustness of the estimated coefficients for other variables and confirms consistency with the baseline regression, thereby reinforcing confidence in the model's reliability and validity.

**Table 5.** Results of the robustness test.

| Variables | (1) Resil_N | (2) Resil_D | (3) Resil |
|---|---|---|---|
| Market | 0.044 * | 0.354 *** | 0.118 *** |
| | (0.024) | (0.080) | (0.041) |
| IR | −0.462 *** | −1.361 *** | −0.856 *** |
| | (0.133) | (0.440) | (0.220) |
| Infor | 0.223 * | 2.677 *** | 0.774 *** |
| | (0.126) | (0.335) | (0.235) |
| Labor | 0.014 | 0.031 | 0.076 *** |
| | (0.028) | (0.032) | (0.028) |
| Illit/Edu | −0.016 * | −0.042 | 0.230 ** |
| | (0.009) | (0.058) | (0.109) |
| Patent | 0.010 *** | 0.025 *** | 0.011 * |
| | (0.004) | (0.009) | (0.006) |
| Gov | 0.097 | 3.177 | 1.951 *** |
| | (0.471) | (4.188) | (0.733) |
| Constant | 0.128 | 7.634 *** | 0.212 |
| | (0.244) | (1.188) | (1.020) |
| Obs | 372 | 372 | 372 |
| $R^2$ | 0.437 | 0.775 | 0.408 |

Note: Robust standard error is reported in parentheses, *** $p < 0.01$, ** $p < 0.05$, * $p < 0.1$.

*4.5. Discussion of Endogeneity*

This section addresses potential endogeneity issues in the model regression, such as omitted variables and bidirectional causality, which could bias the estimation results.

The existing literature predominantly focuses on six key dimensions when examining the determinants of regional economic resilience: industrial structure, innovation capacity, governance efficacy, level of urbanization, human capital, and economic openness. However, the indicators utilized in this study overlook three influential factors: governance efficacy, urbanization level, and economic openness. To bridge this gap, we have incorporated corresponding economic indicators to enhance the model's robustness. Specifically, governance efficacy is measured using the proportion of regional general budget expenditure to GDP (Gov), urbanization level is measured using the ratio of urban population to the total resident population at each province's year-end (Urb), and economic openness is measured using the proportion of total goods imports and exports to GDP (Open) [43]. Regression results from Models (1) to (3) in Table 6 incorporate these additional variables one by one. It is observed that compared to the baseline regression outcomes, the estimated coefficient magnitudes of the influencing factors exhibit slight variations with the inclusion of the supplementary variables. However, crucially, the significance levels and directional indicators remain consistent, affirming the robustness of the baseline model.

**Table 6.** Results of endogeneity problems.

| Variables | (2) Resil | (3) Resil | (4) System GMM |
|---|---|---|---|
| L.resil | | | 0.214 *** |
| | | | (0.023) |
| L2.resil | | | −0.165 *** |
| | | | (0.009) |
| Market | 0.115 ** | 0.102 ** | 0.124 *** |
| | (0.048) | (0.048) | (0.015) |
| IR | −0.865 *** | −0.778 *** | −0.915 *** |
| | (0.245) | (0.250) | (0.157) |
| Infor | 0.760 *** | 0.669 *** | 0.960 *** |
| | (0.228) | (0.235) | (0.167) |
| Labor | 0.073 *** | 0.067 ** | 0.091 *** |
| | (0.027) | (0.028) | (0.012) |
| Illit | −0.040 ** | −0.040 ** | −0.030 * |
| | (0.018) | (0.018) | (0.016) |
| Patent | 0.012 ** | 0.025 *** | 0.033 *** |
| | (0.006) | (0.007) | (0.002) |
| Gov | 1.731 ** | 1.382 * | 0.146 |
| | (0.763) | (0.735) | (0.367) |
| Urb | −0.007 | 0.001 | |
| | (0.012) | (0.012) | |
| Open | | −1.040 *** | |
| | | (0.364) | |
| Constant | −1.887 ** | −1.754 ** | −1.347 *** |
| | (0.774) | (0.775) | (0.354) |
| AR(1) (*p* Value) | | | 0.007 |
| AR(2) (*p* Value) | | | 0.386 |
| Hansen (*p* Value) | | | 0.205 |
| Obs | 372 | 372 | 310 |
| $R^2$ | 0.414 | 0.427 | |

Note: Robust standard error is reported in parentheses, *** $p < 0.01$, ** $p < 0.05$, * $p < 0.1$.

To address potential reverse causality between dependent and independent variables, this study employs the system Generalized Method of Moments (GMM) technique for model testing. This method utilizes one-period and two-period lags of the independent variables as instrumental variables. The regression results are displayed in model (4) of Table 6. The results show that the second-order autocorrelation (AR(2)) is above 0.1, and the Hansen test yields a *p*-value of 0.104. Thus, the null hypothesis is not rejected, suggesting the validity of the instrumental variables used. Crucially, the significance levels and directions of the coefficients for each independent variable remain consistent with those of the benchmark regression, further confirming the robustness of the findings.

*4.6. Heterogeneity Analysis*

4.6.1. Regional Heterogeneity Analysis

This section analyzes regional heterogeneity across the sample. Due to the small sample size, dummy variables are assigned to the eastern, central, western, and northeastern regions to facilitate distinction. Models (1)–(4) in Table 7 are the total regression results of regional heterogeneity, which are divided into four columns for the convenience of analysis. The regression outcomes depicted in Table 7 reveal the significance of the marketization index within the western and northeast regions. This suggests that, in comparison to the eastern and central regions, the level of marketization in the western and northeast regions is relatively lower. Consequently, there exists a greater imperative to advance market-oriented reforms as a means to enhance regional economic resilience in these areas.

**Table 7.** Results of regional heterogeneity.

| | East (1) | Middle (2) | West (3) | Northeast (4) |
|---|---|---|---|---|
| Market × dummy | 0.085 | 0.033 | 0.102 ** | 0.424 * |
| | (0.077) | (0.230) | (0.049) | (0.216) |
| IR × dummy | −1.015 *** | −0.391 | −1.038 *** | −0.625 |
| | (0.344) | (0.636) | (0.377) | (0.512) |
| Infor × dummy | 0.872 *** | 1.248 | 0.934 ** | 1.210 |
| | (0.281) | (1.554) | (0.377) | (1.141) |
| Labor × dummy | 0.007 | 0.101 * | 0.191 *** | 0.489 *** |
| | (0.022) | (0.061) | (0.046) | (0.088) |
| Illit × dummy | −0.090 ** | −0.095 | −0.013 | −0.405 * |
| | (0.035) | (0.066) | (0.021) | (0.214) |
| Patent × dummy | 0.011 * | 0.018 | 0.056 | 0.158 |
| | (0.006) | (0.086) | (0.059) | (0.185) |
| Gov × dummy | 0.012 | −0.239 | 0.119 | −0.914 |
| | (0.164) | (0.536) | (0.184) | (0.825) |
| Constant | −1.872 | −1.872 | −1.872 | −1.872 |
| | (2.010) | (2.010) | (2.010) | (2.010) |
| Obs | 372 | 372 | 372 | 372 |
| $R^2$ | 0.473 | 0.473 | 0.473 | 0.473 |

Note: Robust standard error is reported in parentheses, *** $p < 0.01$, ** $p < 0.05$, * $p < 0.1$.

The shift in industrial structure towards the service sector significantly boosts economic resilience in the eastern and western regions. The central region is primarily characterized by agriculture and raw material processing industries, while the northeast region is predominantly associated with heavy industry. Thus, these regions should prioritize the development of their respective pillar industries while continually fostering the growth of modern service sectors. In contrast, the western region primarily focuses on resource-intensive industries, such as raw material development and processing, leading to excessive resource consumption and environmental degradation. In contrast, the service industry entails lower energy and resource consumption and generates less environmental pollution. Additionally, it stimulates consumption and alleviates employment pressures. Consequently, relative to the central and northeastern regions, the eastern and western regions exhibit greater suitability for fostering the service industry, aligning with local industrial development needs. This strategic emphasis on the service sector serves to optimize regional industrial structure, thereby enhancing regional resilience.

Apart from the central region, the advancement of information technology in the eastern, western, and northeastern regions emerges as a catalyst for enhancing economic resilience. By evaluating the telephone penetration rate (ministry per 100 people) in 2020, it is evident that the eastern region averaged 147, the western region averaged 123, and the northeastern region averaged 131, surpassing the central region's average of 110. These data underscore the comparatively lower level of information technology in the central region, indicating a need for further development in communication infrastructure. Consequently, the potential for enhancing regional economic resilience through information technology improvement in the central region appears limited.

The increase in the labor force significantly enhances the economic resilience of central, western, and northeastern regions. This trend arises because the more developed eastern coastal regions, with their superior public services and infrastructure, attract a significant portion of the labor force from the aforementioned areas. Such migration patterns underscore the uneven distribution of labor resources, which in turn affects regional economic resilience. Furthermore, the quality of the labor force emerges as a pivotal factor influencing the economic resilience of eastern and northeastern regions. First of all, data from the Seventh National Census reveals a national illiteracy rate of 2.67%, with five provinces in the eastern region reporting rates above this national average. This disparity is exacerbated by the considerable influx of migrant workers in the eastern regions, leading to a

heterogeneous educational profile among the labor force. The implications of these findings suggest that while the size of the labor force underpins the economic resilience of less developed regions, the quality of the labor force—encompassed by educational attainment and skills development—is critical for enhancing economic resilience in more developed areas. Secondly, most of the talents trained by universities in the central and western regions flow to the eastern region, resulting in a brain drain, and most of the people left behind are elderly groups with low cultural levels. Even if the illiteracy rate is reduced, it can not significantly improve regional economic resilience. Finally, the illiteracy rate in the three northeastern provinces has always been at a low level. However, combined with the development of local industrial structure, it is speculated that the possible reason is that the northeast region uses CNC machine tools, heavy equipment, iron and steel metallurgy, and petrochemical and other heavy industries as pillar industries, and the transformation and upgrading of the economic structure has a large demand for technical talents. Thus, reducing the illiteracy rate through enhanced educational initiatives and increasing the skill level of the workforce is imperative. By improving the skill level of the labor force and optimizing the allocation of human resources, the region can better withstand external crises.

Additionally, the capacity for innovation markedly strengthens economic resilience, particularly within the eastern region. This observation underscores a notable discrepancy in innovation levels between the eastern region and the other three regions (central, western, and northeastern), with the former showcasing a considerable lead in innovation. As a result, the positive impact of innovation on economic resilience is more pronounced in the eastern region, while it appears more subdued in the central, western, and northeastern regions.

4.6.2. Heterogeneity Analysis Based on Marketization

The preceding analysis underscores the presence of regional heterogeneity in the influence of the marketization index on regional economic resilience. To further investigate the focus direction of promoting marketization reform in each region, this paper adopts five marketization subdivision indexes to replace the total marketization index for regression and sets dummy variables to distinguish the eastern, central, western, and northeast regions. Other explanatory variables were set as control variables, and the results are shown in columns (1)–(4) of the model in Table 8. The table is divided into four columns for ease of analysis.

**Table 8.** Results of marketization heterogeneity.

| Variables | (1) Eastern | (2) Middle | (3) Western | (4) Northeast |
|---|---|---|---|---|
| market1 | 0.031 | −0.084 | −0.017 | 0.237 |
| (The relationship between government and market) | (0.047) | (0.057) | (0.020) | (0.191) |
| market2 | 0.158 | −0.138 | 0.003 | 0.221 *** |
| (The development of the non-state economy) | (0.104) | (0.234) | (0.051) | (0.081) |
| market3 | 0.018 | 0.102 * | 0.026 | 0.091 *** |
| (Product market development degree) | (0.024) | (0.052) | (0.029) | (0.033) |
| market4 | −0.009 | 0.150 ** | 0.067 ** | −0.053 |
| (Factor market development degree) | (0.033) | (0.066) | (0.029) | (0.088) |
| market5 | 0.003 | 0.025 | 0.044 | 0.334 ** |
| (Market intermediary organization development degree and legal environment) | (0.034) | (0.053) | (0.036) | (0.133) |
| Control variable | Yes | Yes | Yes | Yes |
| Fixed effect | Yes | Yes | Yes | Yes |
| Constant | −3.138 *** | −3.138 *** | −3.138 *** | −3.138 *** |
| | (1.129) | (1.129) | (1.129) | (1.129) |
| Obs | 372 | 372 | 372 | 372 |
| R^2 | 0.513 | 0.513 | 0.513 | 0.513 |

Note: Robust standard error is reported in parentheses, *** $p < 0.01$, ** $p < 0.05$, * $p < 0.1$; Columns (1)–(4) are the interaction results of dummy variables and influencing factors in different regions. For the convenience of analysis, they are divided into different groups in this paper.

The findings indicate significant potential for market reform improvements across all regions. In the eastern coastal region, market reform has exhibited decisive development, and the marginal contribution to economic resilience has been negligible. However, in the other three regions, market reform still needs to be carried out through different aspects. According to the evolution law of the regional economic system, the continuous accumulation of capital, labor, and technology within the regional system has propelled rapid development, culminating in the peak of the adaptability cycle for the regional economy. Concurrently, the connectivity between various market entities has undergone continuous enhancement, thereby bolstering the region's capacity to navigate external shocks. However, a notable "barrel effect" exists in the central, western, and northeastern regions, which involves significant challenges.

Although the marketization of the central region has achieved certain results on the whole, it can still improve the product market, especially the institutional construction of the factor market to enhance the local economic resilience. Taking into account the rapid development of private enterprises in the central region, the industrial development environment is poor, and the market-oriented degree of land, capital, talent, and other factors is low, resulting in low-end product circulation and resources are not fully utilized. Therefore, Central China still needs to continuously improve these two aspects to ensure sustainable economic development. In the western region, natural resources are rich and of different quality. However, the industrial chain is short, and the value chain is low. There are many regulation and system deficiencies in resource pricing, and the low degree of marketization leads to the low utilization rate of local factor inputs. Hence, for the western region, prioritizing institutional reforms within the factor market is paramount. Enhancing the efficiency of local natural resource allocation and adopting a strategy of maximizing resource utilization are pivotal steps toward augmenting the region's resilience to external crises. Similarly, Northeast China must enhance its regional economic resilience by promoting the development of the non-state-owned economy. This involves improving the product market, strengthening market intermediary organizations, and enhancing the legal environment. Initiatives should focus on reducing the dominance of state-owned enterprises, which currently suffer from poor performance due to excessive governmental intervention, and on standardizing transactions in product and factor markets by establishing effective market supervision and relevant systems.

*4.7. Discussion*

The concept and theoretical framework of regional economic resilience, emerging as a significant focus within regional economics and economic geography, are subject to rigorous scrutiny. In quantitative analyses, the diversity of research frameworks adopted by scholars leads to variations in the choice of economic indicators, which in turn results in differing measurements of regional economic resilience. Given the developmental imbalances among regions in China, this study tailors its approach by measuring regional economic resilience based on the regional average growth rate. This method offers a more comprehensive depiction of regional responses to shocks compared to methods based solely on the Chinese average. However, whether this improvement can be extended to urban agglomeration, prefecture-level cities, or a broader level and become a measurement concept that can be applied and has a more complete theoretical basis remains unverified.

With the deepening of the research on economic resilience, scholars continue to try to define its concept and put forward measurement methods, and it is these efforts that try to achieve some understanding and recognition for regional economic resilience in theory and policy practice. Therefore, determining the important position of regional economic resilience in macroeconomic development and its positioning in national development planning enhances the relevance and value of understanding regional economic resilience.

## 5. Conclusions and Policy Recommendations

*5.1. Conclusions*

Drawing on relevant literature and theoretical foundations, this paper defines regional economic resilience as the capacity of a regional economic system to effectively respond to external shocks, thereby restoring or enhancing its developmental trajectory. It comprehensively synthesizes measurement methodologies, influential factors, and pertinent theoretical underpinnings associated with regional economic resilience. It also explores internal and external shocks that regional systems face, detailing the varied impacts of these shocks using case analysis methods. Additionally, this paper refines causal prediction within the counterfactual method, measures the economic resilience levels of 31 Chinese provinces, and illustrates the spatial evolution of economic resilience across four distinct periods using ArcGIS. Moreover, through the construction of an econometric model, it empirically tests the influencing factors of regional economic resilience, scrutinizing robustness and addressing endogenous issues. The key findings of this study are as follows: (1) Overall, the economic resilience level of the 31 provinces in China has exhibited significant improvement from 2001 to 2020. Notably, the disparities in economic resilience among regions surpass those observed in economic development levels, with significant intraregional variations. (2) The marketization index, industrial structure, information level, labor force size, labor force quality, innovation ability, and the degree of government intervention all exert discernible impacts on regional economic resilience. Moreover, there are conspicuous regional disparities in the effects of various aspects of marketization on regional economic resilience.

*5.2. Policy Recommendations*

Based on the empirical research conclusions and the current realities of China's regional development, this paper offers several recommendations to enhance regional economic resilience across various regions of China.

First, the primary goal of enhancing regional economic resilience is to prepare for various sudden shocks, each of which can potentially inflict significant harm upon both the economy and society. Presently, there exist marked disparities in the level of economic resilience among regions, underscoring the critical imperative to minimize such regional disparities while concurrently enhancing economic resilience across diverse regions. In this regard, the eastern region, while maintaining its economic leadership, should prioritize the stability of all facets of the economic and social system. It is essential to promptly address any vulnerabilities to prevent the "wooden barrel effect", where the overall resilience is undermined by specific weaknesses. The central and western regions ought to persist in advancing strategies aimed at elevating the central region and fostering the development of the western region. Leveraging their inherent advantages and abundant natural resources, these regions should endeavor to cultivate industries with distinct local characteristics that align more closely with local conditions. The government's facilitation of the orderly transfer of industries from the eastern region to the central and western regions presents a favorable opportunity. Given their well-established infrastructure and robust "soft power", these regions are well-prepared to absorb industrial transfers from the east. The northeast region must undertake a paradigm shift in its traditional production mode, with the revitalization of old industrial bases remaining a paramount objective. Continuous efforts to upgrade the industrial structure are crucial for achieving significant regional economic resilience.

Second, despite significant achievements in market-oriented reform, China's marketization is incomplete, with a stark imbalance in marketization levels across regions. While the eastern coastal provinces have made substantial progress, other regions require comprehensive reforms in various dimensions. In the central and western regions, although the product market shows advanced development, the factor markets—comprising land, capital, and labor—are still underdeveloped and lack effective oversight. To address this, the government must persist in advancing reforms in the factor market and enhancing

institutional norms governing factor market operations. In contrast, Northeast China grapples with a significant presence of state-owned enterprises dominating the macro-economy, imposing stringent regulatory constraints on non-state-owned enterprises to some extent. Therefore, the government should strengthen the entry control and public supervision of monopolistic industries, introduce a market competition mechanism to improve regional productivity and sustainable economic development, and enhance the regional ability to resist external shocks.

Third, regional disparities in industrial structures entail varying demands for both the quantity and types of labor force across regions. Due to information asymmetry and imperfect market systems, optimizing the allocation of labor resources proves challenging. In the eastern coastal areas, particularly in large developed cities, there exists a pronounced concentration of labor force. Conversely, the central, western, and northeastern regions, especially small and medium-sized cities, exhibit fewer employment opportunities and lower levels of employment, rendering them less appealing to high-quality and skilled talents. This imbalance between labor force supply and demand, particularly for high-quality labor, remains a significant concern. Therefore, future strategies should focus on promoting the rational movement of labor and enhancing investments in human capital. Improving labor quality through vocational training and increasing the labor force quantity are essential steps to address the chronic labor supply shortage in China.

Fourth, local administrations ought to prioritize the pivotal role of industrial structure and innovation prowess in fostering economic resilience. By upgrading the industrial landscape, regions can mitigate the risks associated with entrapment in outdated skill sets prevalent in traditional sectors, thereby fortifying their capacity to navigate risks and enhance resilience in the face of a dynamic and evolving external milieu. Moreover, fostering technological linkages across various industries and facilitating the conversion of innovations into tangible outcomes are paramount. These initiatives can mitigate structural vulnerabilities from shocks and accelerate economic recovery after crises. Innovation is a key driver of regional progress; therefore, local authorities should rapidly implement innovation-driven development strategies, enhance local innovation capabilities, and position regional innovation as a central pillar in building economic resilience.

### 5.3. Research Prospects

Given the imbalanced development among regions in China, this study enhances the existing methodologies for measuring regional economic resilience by incorporating regional characteristics. It evaluates the economic resilience of different areas using two approaches: regional average growth rates and regional forecasting models. This method offers a more comprehensive representation of each region's capacity to respond to shocks compared to previous resilience levels calculated based on national averages. However, the applicability of this improved approach to broader contexts, such as urban agglomerations, prefecture-level cities, or more extensive areas, and its establishment as a universally applicable concept with a robust theoretical foundation still needs further exploration.

**Author Contributions:** Conceptualization, C.T.; methodology, C.T.; software, C.T.; validation, C.T. and X.Z.; formal analysis, X.Z.; investigation, C.T.; resources, C.T.; data curation, C.T.; writing—original draft preparation, C.T.; writing—review and editing, C.T. and X.Z.; visualization, C.T.; supervision, X.Z.; project administration, C.T.; funding acquisition, C.T. and X.Z. All authors have read and agreed to the published version of the manuscript.

**Funding:** This research received no external funding.

**Institutional Review Board Statement:** Not applicable.

**Informed Consent Statement:** Not applicable.

**Data Availability Statement:** Data are contained within the article.

**Conflicts of Interest:** The authors declare no conflicts of interest.

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
