# Peer review of "Measurement and Influencing Factors of Regional Economic Resilience in China"

_sustainability, doi:10.3390/su16083338_

Round 1
Reviewer 1 Report
Comments and Suggestions for Authors
This is an interesting paper, which is carefully constructed and analyzed. I have two minor points and one major point for improvement.
Minor points:
1. The paper requires a moderate amount of language editing.
2. There is some notational confusion in the section on the construction of the resilience index. Upper case E is used to describe both expected values and regional or national averages. This should be rectified.
Major point
Resilience is based not only on intrinsic factors related to the economy such as marketization, type of economic activity, quality of the labor force and so on, but also on other stabilizing factors such as government support. Surely, government intervention in the form of policies or simply government expenditure has an impact on economic resilience. This may have no effect if it is equal in all regions, but in some cases it can be vary
So I suggest a short one page discussion on what contributes to resilience in the section on the construction of the resilience index.
I also strongly recommend some measure of government intervention or spending (per capita) as an explanatory variable. If not, a strong case needs to be made for their exclusion.
Comments on the Quality of English LanguageThe English needs to be revised using a language editor. It is not bad, but there are a few errors and some clumsiness.
Author Response
We are grateful for the reviewer's suggestion and have refined the language of the manuscript accordingly.
Thank you for your advice. We have clarified the meaning of the symbol E in two formulas within Section 3.1 of the manuscript.
We appreciate the reviewer's suggestion and have included a review of literature related to government intervention in Chapter 2, along with additional content in subsequent explanations.
Thank you for your suggestion. We have now included the degree of government intervention as one of the explanatory variables, and the empirical results have been updated accordingly.

Reviewer 2 Report
Comments and Suggestions for Authors
Dear authors
In this study, the effects of regional economic resilience are analysed using dynamic panel data for China for the period 2001-2021. The analysis confirms that regional economic resilience has improved in 31 provinces and finds that specific factors contribute to this improvement. The results also show regional heterogeneity. In the wake of the Corona Shock, governments have been working to strengthen their economies. Therefore, the results of this study may have beneficial consequences for those involved in production in developing countries. Here, we would like to suggest some improvements.
The period analysed in this study is 2001-2021. Here, the Corona shock in 2020 caused a major economic shock to the Chinese economy through 2021. This event suggests that structural changes are taking place. Does this study examine or verify any specifics with regard to this structural change? It may also be worth explaining how the Corona shock affects the results obtained in this study.
It is common in literature reviews in empirical analysis to analyse the hypothetical propositions that they are trying to clarify. This study should follow suit.
As for how to cite references, follow the method specified by the journal.
Sincerely.
Author Response
We are thankful for the reviewer's recommendation and have added a description of the impact of the COVID-19 pandemic on the economic resilience of various regions in China during 2020 to 2021 in Section 4.1.
Thank you for your suggestion. We have analyzed the hypothetical propositions in the literature review of Chapter 2 and proposed the research hypothesis of this paper.

Reviewer 3 Report
Comments and Suggestions for Authors
The paper investigates an interesting topic such as Measurement and influencing factors of regional economic resilience in China. The methodology is pertinent and English is also good. However there are some issues that need to be addressed.
Introduction The novelty against the existing literature needs to be discussed in order to support the originality of the study. The role of community resilience needs to be discussed in the background of resilience and the due literature review needs to be cited.
Section 2. The existing methodologies to assess resilience of systems (buildings, infrastructures, communities, systems) needs to be discussed. Please refer to; https://doi.org/10.1016/j.istruc.2023.105015 https://doi.org/10.1108/09653560010335068
Section 3 Also, a discussion on the mentioned approaches (such as methods of counterfactual forecasting, statistical time series model or some appropriate structural model) is necessary.
Section 4 The calculations that are at the base of Figure 1 are necessary and not well described. Please expand this part. Comparing the results with existing literature is also necessary. The authors wrote that the regression coefficient of informatization level is 0.900, and the significance level test of 1% indicates that the improvement of informatization level can significantly enhance the regional economic resilience. This needs to be discussed by comparing with other examples and applications.
Conclusion This part needs to focus on three main aspects: limitations of the study, possible applications and future work. The other issues need to be discussed in the previous section. Please reorganize them.
Author Response
We are grateful for your advice. In the second paragraph of Chapter 1, we have added the significant role of regional economic resilience and supported it with relevant literature. In the third paragraph, we have highlighted the gaps in current literature to contrast with the novelty of this study.
Thank you for the suggestion. We have added the two mentioned articles in Chapter 2's literature review and evaluated the existing methods for assessing systemic resilience.
We appreciate the reviewer's suggestion and have included descriptions of statistical time series models and structural model methods in Section 3.1.
Thank you for your advice. In Section 4.1, we have expanded the computational description of Figure 1 and added comparisons with results from related literature. Section 4.3 now includes additional examples to illustrate how improvements in the level of informatization can significantly enhance regional economic resilience.
We are thankful for the suggestion. In Section 5.2, we have described the potential applications of our research findings and added Section 5.3 to outline future research prospects.

Reviewer 4 Report
Comments and Suggestions for Authors
The article is dedicated to the important and relevant topic, shows a high level of sophisticated spatio-temporal analysis and substantial efforts of authors regarding the data collection, processing, visualization, and results interpretation.
Being rather complicated methodologically, the article shows well developed data-driven conclusions about spatio-temporal shifts in economic resilience within 31 provinces in China.
There are some suggestions and discussion points to consider for the reviewed article:
-
Literature review is rather descriptive and the conclusion about the necessity of more granular geo levels for China is not connected with the literature analysis results.
-
The reasoning for selection of the independent variables can be outlined more precisely. The way how the variables are calculated is shown very well, but the question why they were selected is not answered in the section 3 Methods and Data. The authors touch upon this topic a bit in the subsection 4.6 Discussion of endogeneity, but it seems more logical to be covered in the section 3.
-
Despite the combination of resistance period and recovery period into a formula of economic resilience (lines 143 - 146) not diving deeper into the components, the results are presented separately by resistance and recovery with the scale for resilience (Figure 1 and Figure 2). This contradiction might be confusing for the readers and limits the potential of the research results interpretation by the journal audience. The way of the results presentation here is recommended for reevaluation.
-
From the methodological POV it is worth considering comparison of the results to the major economic and social trends not only for crisis and post crisis time frame, but relatively stable periods. E.g., taking into account the provinces development trajectories for the whole study timeframe 2001 - 2021.
Author Response
Thank you for the reviewer's suggestion. We have enriched the existing literature study in Chapter 2 and added the reasons for selecting various control variables in Section 3.3.2.
We appreciate the reviewer's comment. The division and definition of resistance and recovery periods aim to better illustrate the economic performance of different regions in China over various years. The resilience values calculated demonstrate the economic performance of these regions, and we have made efforts to clarify this stance in the empirical analysis explanation.
Thank you for your comment. We have added specific conditions of certain provinces and cities in Section 4.1.

Reviewer 5 Report
Comments and Suggestions for Authors
The abstract should be extended with recommendations and information on further research steps.
Keywords should be corrected to avoid repeatable words in the title of the manuscript. For example, instead of "regional economic resilience" authors can use "national economy" and "sustainable development".
In the "References" chapter, the five years older sources dominate (about 70%). A suggestion to the authors is to explore more recent research in this area published in journals ranked Q1/Q2.
The manuscript contributes to understanding regional economic resilience in China and provides concrete guidelines for policy actions that could enhance this resilience in the future.
The manuscript contains research on the importance of regional economic resilience and its ability to adapt and withstand external crises, highlighting its key role in the sustainable development of the national economy.
Using the chosen methodology, the authors measured the economic resilience index of 31 localities in China for two decades, including two pandemic years, and conducted an empirical analysis of factors affecting regional economic resilience.
The research indicates a significant improvement in economic resilience in China during the observed period, with differences in resilience between regions significantly greater than differences in the economic development level. Also, the research provides policy recommendations to further improve regional economic resilience in the different China regions.
The manuscript contains a usable model for understanding the regional economic resilience of other countries by the academic and business development community.
Author Response
We are grateful for the reviewer's suggestion and have expanded the policy recommendations in the abstract.
Thank you for the invaluable suggestion. We have corrected the keywords to "national economy" and "sustainable development" as advised.
Thank you for the suggestion. We have incorporated articles related to our topic published in Q1 and Q2 journals over the past five years into the references.

Round 2
Reviewer 1 Report
Comments and Suggestions for Authors
The paper is now considerably improved. The suggestions about incorporating government intervention have been taken up. The arguments in the paper are now very convincing. Some minor language is desirable
Comments on the Quality of English LanguageMinor language editing is desirable
Author Response
Thanks for your suggestion, we have polished the article (marked red in the picture).

Reviewer 2 Report
Comments and Suggestions for Authors
Dear author.
No further improvements seem necessary. Good luck.
Sincerely.
Author Response
Thank you very much.
We wish you all the best.

Reviewer 3 Report
Comments and Suggestions for Authors
The paper is ready to be accepted
Author Response

(The authors gave the same response as above.)
